# How to Improve Healthcare for Patients with Multimorbidity and Polypharmacy in Primary Care: A Pragmatic Cluster-Randomized Clinical Trial of the MULTIPAP Intervention

**DOI:** 10.3390/jpm12050752

**Published:** 2022-05-06

**Authors:** Isabel del Cura-González, Juan A. López-Rodríguez, Francisca Leiva-Fernández, Antonio Gimeno-Miguel, Beatriz Poblador-Plou, Fernando López-Verde, Cristina Lozano-Hernández, Victoria Pico-Soler, Mª Josefa Bujalance-Zafra, Luis A. Gimeno-Feliu, Mercedes Aza-Pascual-Salcedo, Marisa Rogero-Blanco, Francisca González-Rubio, Francisca García-de-Blas, Elena Polentinos-Castro, Teresa Sanz-Cuesta, Marcos Castillo-Jimena, Marcos Alonso-García, Amaia Calderón-Larrañaga, José M. Valderas, Alessandra Marengoni, Christiane Muth, Juan Daniel Prados-Torres, Alexandra Prados-Torres

**Affiliations:** 1Research Unit, Primary Care Assistance Management, Madrid Health Service (Servicio Madrileño de Salud), 28035 Madrid, Spain; juanantonio.lopez@salud.madrid.org (J.A.L.-R.); cristinamaria.lozano@salud.madrid.org (C.L.-H.); elena.polentinos@salud.madrid.org (E.P.-C.); teresa.sanzcu@salud.madrid.org (T.S.-C.); 2Department of Medical Specialties and Public Health, School of Health Sciences, Rey Juan Carlos University (Universidad Rey Juan Carlos), 28922 Madrid, Spain; 3Research Network on Health Services in Chronic Diseases, Red de Investigación en Servicios de Salud en Enfermedades Crónicas (REDISSEC)-Red de Investigación Cooperativa Orientadas a Resultados en Salud (RICORS-RICAPPS), ISCIII, 28029 Madrid, Spain; francisca.leiva.sspa@juntadeandalucia.es (F.L.-F.); agimenomi.iacs@aragon.es (A.G.-M.); bpoblador.iacs@aragon.es (B.P.-P.); vpico@salud.aragon.es (V.P.-S.); lugifel@gmail.com (L.A.G.-F.); maza@salud.aragon.es (M.A.-P.-S.); mariaeloisa.rogero@salud.madrid.org (M.R.-B.); franciscagonzalezrubio@gmail.com (F.G.-R.); fgblas@salud.madrid.org (F.G.-d.-B.); amaia.calderon.larranaga@ki.se (A.C.-L.); sprados.iacs@aragon.es (A.P.-T.); 4Instituto de Investigación Sanitaria Gregorio Marañon IISGM, 28009 Madrid, Spain; 5Ricardos General Health Center (Centro de Salud General Ricardos), 28019 Madrid, Spain; 6Multiprofessional Family and Community Care Teaching Unit (Unidad Docente Multiprofesional de Atención Familiar y Comunitaria) of the Málaga-Guadalhorce Health District, 29016 Málaga, Spain; juand.prados.sspa@juntadeandalucia.es; 7Biomedical Research Institute of Malaga (Instituto de Investigación Biomédica de Málaga-IBIMA), 29010 Málaga, Spain; 8Department of Pharmacology and Paediatrics, School of Medicine, University of Malaga (Universidad de Málaga), 29016 Málaga, Spain; 9EpiChron Research Group, Aragon Health Sciences Institute (IACS), IIS Aragón, Miguel Servet University Hospital, 50009 Zaragoza, Spain; 10Las Delicias Health Center (Centro de Salud las Delicias), Málaga-Guadalhorce Health District, 29016 Málaga, Spain; fernando.lopez.verde.sspa@juntadeandalucia.es; 11Primary Health Care Research and Innovation Foundation (Fundación Investigación e Innovación Biosanitaria en Atención Primaria), FIIBAP, 28003 Madrid, Spain; 12Torrero-La Paz Primary Care Health Centre, Aragon Health Service (SALUD), 50007 Zaragoza, Spain; 13La Victoria Health Center (Centro de Salud la Victoria), Málaga-Guadalhorce Health District, 29013 Málaga, Spain; josefa.bujalance.sspa@juntadeandalucia.es; 14San Pablo Primary Care Health Centre, Aragon Health Service (SALUD), School of Medicine, University of Zaragoza, 50009 Zaragoza, Spain; 15Primary Care Pharmacy Service Zaragoza III, Aragon Health Service (SALUD), 50009 Zaragoza, Spain; 16Delicias Sur Primary Care Health Centre, Aragon Health Service (SALUD), 50009 Zaragoza, Spain; 17Dr Mendiguchia Carriche Health Center (Centro de Salud Dr Mendiguchia Carriche), 28914 Madrid, Spain; 18Coín Health Center (Centro de Salud de Coín), Málaga-Guadalhorce Health District, 29100 Málaga, Spain; marcos.castillo.sspa@juntadeandalucia.es; 19Preventive Medicine Unit, University Hospital Alcorcón Foundation (Hospital Universitario Fundación Alcorcón), 28922 Madrid, Spain; marcos.alonso@madrid.org; 20Aging Research Center, Department of Neurobiology, Care Sciences and Society, Karolinska Institutet and Stockholm University, 17165 Stockholm, Sweden; 21Department of Family Medicine, National University Health System and Department of Medicine National University of Singapore, Singapore 119228, Singapore; jmvalderas@nus.edu.sg; 22Department of Clinical and Experimental Sciences, University of Brescia, 25123 Brescia, Italy; alessandra.marengoni@unibs.it; 23Institute of General Practice, Johann Wolfgang Goethe University, 60323 Frankfurt am Main, Germany; christiane.muth@uni-bielefeld.de; 24Department of General Practice and Family Medicine, Medical Faculty OWL, University of Bielefeld, 33615 Bielefeld, Germany

**Keywords:** multimorbidity, polypharmacy, Ariadne principles

## Abstract

(1) Purpose: To investigate a complex MULTIPAP intervention that implements the Ariadne principles in a primary care population of young-elderly patients with multimorbidity and polypharmacy and to evaluate its effectiveness for improving the appropriateness of prescriptions. (2) Methods: A pragmatic cluster-randomized clinical trial was conducted involving 38 family practices in Spain. Patients aged 65–74 years with multimorbidity and polypharmacy were recruited. Family physicians (FPs) were randomly allocated to continue usual care or to provide the MULTIPAP intervention based on the Ariadne principles with two components: FP training (eMULTIPAP) and FP patient interviews. The primary outcome was the appropriateness of prescribing, measured as the between-group difference in the mean Medication Appropriateness Index (MAI) score change from the baseline to the 6-month follow-up. The secondary outcomes were quality of life (EQ-5D-5 L), patient perceptions of shared decision making (collaboRATE), use of health services, treatment adherence, and incidence of drug adverse events (all at 1 year), using multi-level regression models, with FP as a random effect. (3) Results: We recruited 117 FPs and 593 of their patients. In the intention-to-treat analysis, the between-group difference for the mean MAI score change after a 6-month follow-up was −2.42 (95% CI from −4.27 to −0.59) and, between baseline and a 12-month follow-up was −3.40 (95% CI from −5.45 to −1.34). There were no significant differences in any other secondary outcomes. (4) Conclusions: The MULTIPAP intervention improved medication appropriateness sustainably over the follow-up time. The small magnitude of the effect, however, advises caution in the interpretation of the results given the paucity of evidence for the clinical benefit of the observed change in the MAI. Trial registration: Clinicaltrials.gov NCT02866799.

## 1. Introduction

Multimorbidity is an increasingly common phenomenon worldwide [1,2]. In Spain, approximately 37.5% of the population has multimorbidity, 10.6% suffer from ≥5 chronic diseases, and the average number of chronic problems in older adult (65–74 years) is 2.8 [3,4]. The potential negative health impacts of multimorbidity include polymedication, reduced quality of life and functional capacity, increased use of health services, as well as increased complications and health costs [5,6,7]. Polymedication or polypharmacy is present in 20% of the elderly population treated in primary care and up to 37% of the elderly population in general; those aged over 65 taking an average of 4.5–8 drugs/day. It is considered to be the main cause of potentially inappropriate prescribing, which includes excessive, incorrect, and insufficient prescriptions [8] and increases the risk of poor treatment adherence and adverse drug interactions and reactions [9], leading to increased risks of hospitalizations, fractures, morbidity [10], and even mortality [11]. Spain has a well-established framework to address polymedication in patients aged 75 and over, but the population under this age is not subject to specific control or prevention strategies regarding polymedication, although older adults have relatively high rates of multimorbidity and polymedication. Several methods to quantify and reduce potentially inappropriate prescribing have been proposed: explicit measures such as the BEERS and STOPP/START criteria, and implicit measures based on the physician’s clinical judgement that consider a patient’s overall situation, the Medication Appropriateness Index (MAI) [12,13] being the most accepted.

To improve polypharmacy, organizational interventions (use of electronic medical records, feedback to reduce drug interactions, and continuous review of medications), interventions focused on professionals (educational programs for prescribers or consumers), interventions in the care process (help with the decision-making system), interventions in patients (education on the use of medications and treatment objectives), and financial interventions (incentives linked to indicators and regulatory interventions) [14] have been proposed. The systematic review by Rankin et al. concluded that pharmacist-led complex organizational interventions that evaluated pharmaceutical care in different care settings achieved an overall reduction in potentially inappropriate prescribing, but the effects on other variables, such as hospital admissions and quality of life, were conflicting [15]. In 2018, Muth et al. [16] published the results of the PRIMUM study, which evaluated the effectiveness of an intervention to improve appropriate prescribing using the MAI as the main outcome variable, but it did not obtain significant effects.

The Cochrane Systematic Review 2016 conducted by Smith et al. [17], concluded, with moderate strength of evidence, that most interventions yielded a minor improvement in clinical results and in patient-reported results, and one of the interventions reduced mortality at the 4-year follow-up. The evidence supporting effects on the use of health services, treatment adherence, and quality of prescription was weak, and there was a very limited difference in costs. Subsequently, Salisbury et al. [18] evaluated the effectiveness of a patient-focused intervention (3D study) based on the dimensions of health, depression, and drugs, measuring health-related quality of life as the main outcome, without obtaining significant changes. However, several studies (including a 3D study) have shown improved patient experience in individualized care [2,15,18].

The Ariadne principles, developed by consensus by a multidisciplinary group of experts, propose the agreement of realistic therapeutic goals between the physician and the multimorbidity and polypharmacy patient, taking their preferences and desires into account and ensuring their individualized care management and monitoring [19,20]. The core of the Ariadne principles is based on (i) the evaluation of the interaction, that is, the recording of the possible relationships between the patient’s health problems and their therapies, their physical and mental state, and their living environment; (ii) prioritizing health problems and obtaining patient preferences, that is, asking about the most important positive and negative results of the treatment for the patient; (iii) and determination of individualized care to achieve negotiated treatment goals. Their feasibility and impact on primary care have not been evaluated, although the potential benefit of implementing such a strategy in clinical practice of primary care has been recognized [21,22,23]. The primary objective of this study was to evaluate the effectiveness of a complex intervention under the primary care MULTIPAP model based on the implementation of the Ariadne principles on improving medication appropriateness in an elderly population. The secondary objectives were to evaluate the effects of the MULTIPAP intervention on other variables, such as quality of life, use of health services, adherence to treatment, and medication safety.

## 2. Materials and Methods

### 2.1. Methods

We conducted a pragmatic, cluster-randomized, controlled trial in primary care centers in Spain. The Spanish National Health System provided first-contact, comprehensive, continuous, coordinated care (which is free at the point of care) to define a population served by primary care centers. Patients provided the names of family physicians (FPs) who were responsible for delivering and coordinating their care.

We recruited FPs in three regions (Aragon, Madrid, and Andalusia), covering about a third of the Spanish population. The eligible physicians used the electronic health record system, had been in their current position for at least 1 year, and had no prospects of leaving their position during the study.

Eligible patients were aged 65–74 years older, with at least three different chronic conditions and polypharmacy, defined as ≥5 drugs prescribed over the 3 months before inclusion in the study, and had made at least one visit to the FP in the past year. We used the electronic health record system of primary care to identify patients with three chronic diseases, as per O’Halloran’s list [24]. Patients were excluded if they had a life expectancy of less than 12 months, were institutionalized, or suffered from mental and/or physical conditions considered by the FP in order to follow study requirements. Participating patients and their physicians provided written informed consent. The CONSORT checklist is available as Appendix A.

The study was approved by the Ethics Committee for Clinical Research of Aragon (CEICA), was favorably evaluated by the Research Ethics Committee of the Province of Malaga on 25 September 2015 and by the Central Committee of Primary Care Research of the Community of Madrid, and followed the published protocol [25] (Appendix A).

### 2.2. Recruitment

Patients were recruited from December 2016 to January 2017. Voluntary participation was requested of FPs working in primary care health centers in each of the three regions. Patients registered with participating FPs and fulfilling the inclusion criteria were listed at random as potential participants. Each FP consecutively selected five patients from this list and invited them to participate. The FP provided the patient with detailed information about the study, they confirmed the patient’s eligibility and obtained the patient’s written informed consent. Relevant data on invited patients who declined were collected (age, sex, and reason for non-participation). The FPs submitted their own self-reported sociodemographic and multimorbidity training variables before the start of the study. Participant data were collected by the recruiting FPs, who were also responsible for patient follow-up and completion of an ad hoc case report form accessed from their personal computers via the project website using a personal identification code. Three visits were defined for patient data collection: baseline (T0), 6-month follow-up (T1), and 12-month follow-up (T2) (see flow chart in Figure 1).

Strategies to improve the protocol adherence of FPs were implemented, including individual follow-up on the protocol’s milestones and queries, as well as incentives such as messages of appreciation via e-mail, acknowledgement of their contribution through an invitation to co-author scientific reports, and continuous professional development certified training sessions.

### 2.3. Randomization and Masking

Once all the participating FPs (allocation unit) had selected their patients (analysis unit) given their consent, and completed baseline patient-related measures, simple FP randomization was performed centrally by the Madrid Primary Care Research Unit, using the software Epidat 4 (SERGAS, Santiago de Compostela, A Coruña, Spain) and with the “balanced group” option to ensure an equal number of FPs in each group. Afterwards, each FP received the information on the study group assigned and all patients recruited by a FP were included in that FP’s group. FPs were aware of their treatment allocation. All analyses were performed by the trial statisticians, who were blinded to the group assignment.

### 2.4. Intervention

FPs in the control group continued to provide usual care. Usual care is usually provided based on recommendations from the clinical practice guidelines and protocols corresponding to each separate patient chronic disease. Participants in the intervention group received the MULTIPAP complex intervention, which is based on a patient-centered care model, the Ariadne principles. The MULTIPAP intervention was developed in accordance with the recommendations and taxonomy proposed by the Cochrane Effective Practice and Organization of Care Review Group and described in detail in the protocol following the approach proposed by Perera et al. [25,26]. A template for the intervention description and replication (TIDieR) is included in the checementary Appendix A.

The FP training was based on the completion of eMULTIPAP, a 4-week course, using the online platform Modular Object-Oriented Dynamic Learning Environment (MOODLE), (Martin Dougiamas, GNU General Public License. http://www2.iavante.es/es/detalle-curso/2278 (accessed on 1 February2017)) and based on constructivism and problem-based learning. The content was designed by researchers working on the MULTIPAP project and included: multimorbidity, polypharmacy, appropriateness of prescribing, treatment adherence, the Ariadne principles, therapeutic cascade, deprescription and physician–patient shared decision-making basic concepts. The eMULTIPAP course has been assessed according to the Kirkpatrick model and has shown knowledge improvement and high applicability of learning with more motivation to consider multimorbidity in the clinical practice, addressing Lewis’ proposed curriculum for multimorbidity [27,28]. According to Mills [29], a type 4 logic model for this intervention was developed. This logic model is the most appropriate to represent an intervention that can change each time it is applied, depending on the interaction between the different components of the intervention (i.e., the eMULTIPAP course and a patient-centered interview) and the individual elements of the context in which it is applied (i.e., the characteristics of the patient, the healthcare professional, the team, the health center, the health organization, and the epidemiological situation).

After the training, only one appointment was mandatory; each intervention FP conducted a structured physician–patient interview, comprising a review of the treatment plan, inclusion of patient preferences, and a pharmacological treatment plan. The patients were given a printed copy of the plan and follow-up visits by nurses were according to usual care and protocols.

### 2.5. Outcomes

The primary outcome was the appropriateness of prescribing, measured as the between-group difference in the mean MAI score change from the baseline to 6-month follow-up. The MAI rates each medication along ten criteria (indication, effectiveness, correct dosage, correct directions, practical directions, clinically significant drug-drug interactions, clinically significant drug-disease interactions, duplications, correct duration and cost). For each criterion, the index has operational definitions, explicit instructions, and examples, and the evaluator has to rate whether the particular medication is “appropriate”, “marginally appropriate” or “inappropriate”. A MAI medication score was derived by applying weights to each “inappropriate” rating for individual criteria and summing across the 10 individual criteria. Medication scores could range from 0 (no degree of inappropriateness) to 18 (maximum degree of inappropriateness). A final patient summated MAI score was then calculated by summing all medication MAI scores [30,31]. Prescribing appropriateness was assessed by three blinded research family physicians with pharmacological expertise trained for the purpose of the study. These evaluators had access to full patient medical records. To ensure consistent ratings, inter-observer concordance was calculated through the intraclass correlation coefficient (ICC). The ICC started at 0.41 (95% CI 0.11–0.70). Then, the evaluators were trained, after which their ICC reached 0.69 (95% CI 0.31–0.80). Additionally, another FP and a pharmacist different from the MAI evaluators conducted a second appraisal of the inter-observer reliability with a randomly selected 10% of the completed MAIs, ICC 0.78 (95% CI 0.53–0.88). This approach has been proposed in other studies using the MAI [32,33].

As a secondary outcome, the appropriateness of prescribing measured as the between-group difference in the mean MAI score change from the baseline to the 12-month was determined. We collected data about secondary outcomes in five domains: quality of life, patient perception of shared decision making, medication safety, treatment adherence, and use of health services. To assess the quality of life, we used the EuroQol 5D-5 L questionnaire. EQ-5D-5 L is a validated generic instrument that measures health-related quality of life on a visual analogue scale (range from 0 to 100, higher ratings indicate higher quality of life) and five dimensions about mobility, self-care, usual activities, pain/discomfort, and anxiety/depression. Health states are converted into a weighted health state index (full health has a value of 1 and dead a value of 0). We collected data on the five dimensions and the health state index with the method proposed for our country [34].

We assessed patient perceptions of shared decision making using the collaboRATE measure and question number 33 from the National Health System (NHS) inpatient survey [35]. Medication safety was measured as the incidence (number of events per patient year) of adverse drug reactions reported by the FP and potentially hazardous interactions using the taxonomy proposed by Otero-López [36]. Medication adherence was measured with the Morisky Medication Adherence score [37]. Use of health services was measured as unplanned and/or number of hospitalizations, number of visits to emergency services, and number of FP and primary care nurse visits.

Outcomes were collected at baseline, 6-, and 12-month follow-ups, by the FPs. The total time frame for the secondary outcomes was set at 12 months given the characteristics of the variables and the interest in assessing them over a longer period of time. Data were collected during consultations with the FPs. Chronic diseases were coded based on the International Classification for Primary Care and drugs according to the Anatomical Therapeutic Chemical Classification System as reported and reviewed by their FPs. The FPs also reported the full details of deaths in both groups, including date, cause, expectedness, any recent changes in the management of chronic diseases, and any possible association with the intervention. All death and adverse drug reactions reported were reviewed by the Data Sharing Comittee to monitor the trial progress.

### 2.6. Sample Size

The sample size was calculated in order to detect a minimum 2-point between-group difference in the mean MAI score change at 6-month follow-up, assuming a standard deviation of 6 units [15,16] with 80% power, a t2-sided α of 0.05, and accounting for clustering of FP. Although the 2-point difference has been considered clinically relevant in different studies, there is no consensus on a clinical threshold for a specific change in MAI score. Randomized controlled trials (RCTs) using MAI mostly report the mean change in the MAI scores and compare mean differences between groups as a measure of effect size. The calculated sample size allowed us to detect an effect size (d) of 0.33 (2/6), somewhat larger than that described as a small effect by Cohen (d = 0.2) [16,38,39]. Assuming five eligible patients per FP cluster, an intraclass correlation coefficient of 0.03 [40], and a loss-to-follow-up rate of 20%, we needed to recruit 80 FPs and 400 patients (200 per group).

### 2.7. Statistical Analysis

All analyses were carried out in accordance with the intention-to-treat principle, with a prespecified analysis plan for all outcomes [25], using Stata (StataCorp LLC, v14, College Station, TX, USA). We analyzed the impact of the intervention on all outcomes using multi-level regression models (using linear, logistic, or Poisson regression, as appropriate), which included adjustment for baseline measures of the outcome, as well as minimization variables, with FP as a random effect.

Missing data were analyzed using the last observation carried forward (LOCF), as originally proposed in the protocol, as well as through multiple imputation by chained equations, including baseline data. The effectiveness of the intervention on the primary outcome was analyzed using a multiple imputed t-test estimation of the between-group difference in the T1–T0 MAI score changes, with its corresponding 95% CI. A sensitivity analysis was performed to compare the output from both imputation sets. Factors associated with the difference in means were analyzed using a multilevel mixed-effects linear regression model with FP set as a random effect.

As part of the secondary analysis (non-confirmatory) measurements, we analyzed the between-group difference in the T2–T0 MAI score changes using the same method as applied to the main outcome variable (difference in MAI ≥ 2).

### 2.8. Role of the Funding Source

The funder of the study had no role in study design, data collection, data analysis, data interpretation, or writing of the report. The corresponding author had full access to all the data in the study and had final responsibility for the decision to submit for publication.

## 3. Results

### 3.1. Characteristics of the Study Participants

We recruited 118 FPs, and only one of them refused to participate. Out of 636 potentially eligible patients who had been invited to participate, 593 patients accepted, provided informed consent, and were enrolled in the trial. Participating patients had similar characteristics to non-participants (Appendix A). FPs were cluster-randomized to the intervention group (59 FPs, 298 patients) and control group (58 FPs, 295 patients). The follow-up measurements at 6 months were obtained for 570 (96%) participants and at 12 months for 546 (92%) participants (Figure 1).

The average age was 69.7 (2.7), and 55.8% were women. The median number of diseases was 5.0 (interquartile range (IQR) 4.0–7.0). The median (IQR) number of drugs was 7.0 (6.0–9.0). The mean (SD) and median (IQR) baseline summated MAI score was 17.5 (16.8) and 14.0 (5.0–25.0), respectively (Table 1). No relevant differences were found between study arms at baseline in demographic or clinical characteristics or in the study’s primary or secondary outcomes.

### 3.2. Primary Outcomes

At the 6-month follow-up, in our intention-to-treat analysis with multiple imputation, there was a difference between groups in the primary outcome difference in MAI score change (difference in means −2.42 (from −4.27 to −0.59), *p* = 0.009) (Table 2). The sensitivity analyses of the primary outcome at 6 months with the different imputation methods can be found in the Appendix A. In the intention-to-treat analysis, the factors associated with a negative difference in means were receiving the intervention and having good social support. Living alone and taking more medications had the opposite effect on this change in means (Table 3). However, it should be noted that the power for subgroup analysis is limited, no evidence was found of a differential effect in any of our predefined subgroup analyses at 6 months, except for physicians being postgraduate tutors, who showed a greater difference in the MAI score change (Appendix A).

### 3.3. Secondary Outcomes

At the 12-month follow-up, the difference in MAI score change was even higher in the MULTIPAP group, with an adjusted difference in means of −3.40 (95% CI from −5.45 to −1.34), *p* = 0.001. The difference in EQ-5D-5 L (index) scores was −0.006 (95% CI from −0.034 to 0.022) at 12 months.

There was also no evidence that the intervention reduced the number of adverse drug events or improved medication adherence rates. The strength of the main outcome considered ten different aspects related to appropriateness changes in duplication, effectiveness, dosage, and interactions criteria were observed in both groups. In the intervention group, patient’s percentage with an inappropriate MAI criterion in at least one medication at the 6-month follow-up statistically significantly decreased in indication −8.4(95% CI from −16.5 to −0.2), effectiveness −8.7 (95% CI from −16.5 to −0.9), and duplication −16 (95% CI from −22.6 to −9.5). In the control group, only duplication was statistically significance −7.41(95% CI from −13.5 to −1.3).

Patients in the intervention group had more nurse consultations (0.168, 95% CI from −0.08 to 0.41) and more primary-care physician consultations (0.073, 95% CI from −0.10 to 0.25) over 6 months than the usual-care group (Appendix A). This difference was not significant at 12 months. There was no evidence of a difference in the number of hospital admissions or outpatient visits.

As assessed by the shared decision-making tool collaboRATE, both groups presented scores from 7 to 10 in 94% of cases. Regarding Question 33 of the in-patient survey of the NHS, “Were you involved as much as you wanted to be in decisions about your care and treatment?”, the percentage of “Yes, definitely” responses in both groups was approximately 79%, with no differences between groups at either 6- or 12-month follow-ups.

During the trial, four of the 593 patients died, including two patients in the intervention group and two in the usual-care group. None of the deaths was reported as possibly related to the intervention.

## 4. Discussion

The MULTIPAP intervention, based on the Ariadne principles, which includes international guidelines for the treatment of patients with multimorbidity [1,2], is effective in improving the appropriateness of medication at 6- and 12-month follow-ups. Sustained changes in MAI were observed and confirmed across different sensitivity analyses, with no significant statistical or clinical changes in any of the other variables analyzed; both groups showed significant improvements in measures of patient-centered care.

The MAI is considered to be the most reliable and valid implicit instrument to measure the appropriateness of a patient’s medication regime. The studies collected in Hanlon et al. [39] demonstrated both validity and its association with relevant and specific service, in line with the results of the study by Müller et al. [41]. In 1997, Schamder et al. [42] found that a difference of two points was associated with differences in hospitalization and unscheduled outpatient or emergency visits. Several studies reviewing MAI interpretation from the 1990s [30] to the present [16,17] proposed to consider a two–three point change as indicative of a clinical difference [36,38].

There are several ways to interpret the MAI results in studies evaluating the effectiveness of interventions to reduce inappropriate medication [38]. One of them is to compare the summated MAI score post intervention, adjusted by the MAI baseline value [43]. The other way, which is the most commonly used in RCTs, also applied here, is to calculate the mean MAI change and compare the differences in means between groups as a measure of effect size [44]. The five studies included in the 2018 Cochrane review [15] that used this approach found a mean difference of −4.76 (95% CI from −9.20 to −0.33), although in the sensitivity analysis, the mean difference was −0.50 (95% CI from −2.27 to 1.28). The differences in our study were −2.42 (95% CI from −4.27 to −0.59) at 6-month follow-up and −3.40 (95% CI from −5.45 to −1.34) at 12-month follow-up, with the size of the effect increasing over time. This change is comparable to, but less than, that observed in the RCT by Romskaug et al. 2020, with and MAI change of −6.9 (95% CI from −9.1 to -4.7) but with their patients having a mean age of 83.3 (7.3) in contrast to our study mean age of 69.7 (2.7) [45]. It could be possible that some of the observed benefits are due to regression to the mean. If we standardize effect sizes we observe that the effect size was d = 0.28 (CI 95% from 0.11 to 0.44) at 6-month follow-up and d = 0.31 (CI 95% from 0.14 to 0.48) at 12-month follow-up, intermediate values between the small effect size d = 0.20 and moderate d = 0.5 considered by Cohen [44].

The RCT PRIMUM [16] conducted in 72 German general medicine clinics found no differences in the appropriateness of prescription. The mean baseline score on the MAI was 17.5 (16.8) in MULTIPAP as compared with 4.8 (5.4) in RCT PRIMUM. In their trial, the appropriateness of prescription was evaluated by pharmacists reviewing the documentation previously prepared by clinical assistants, unlike our study, in which the evaluators were family physicians with clinical experience and training in polypharmacy who had the medical history of each patient available to them, which could have influenced the MAI score. The baseline MAI score found in MULTIPAP is similar to that of other studies, which have had baseline MAI scores of approximately 14 points [15,17,38,43,44,45,46,47].

None of the intermediate variables seems to have been affected by the intervention, either quality of life or patient-shared decision-making perception. These results are consistent with the PRIMUM study [16], a 3D study [18], and a previous systematic review. Given the complexity of multimorbidity and the impact of the health system on the health of the population, significant differences in quality of life are probably not expected with this type of intervention. Tinetti et al. study’s findings suggested that older adults with multimorbidity priorities care may be associated with reduced treatment burden and unwanted health care [48]. Supporting the prioritization of patient health problems is not an easy task, and it must be taken into consideration that patient preferences change over time [49].

There is wide variability in the methods of measuring and comparing the use of services by multimorbidity patients [23,50,51]. The results at the 12-month follow-up in terms of medical and nursing consultations are similar to those described elsewhere in primary care in Spain [52,53]. There were no relevant changes in either the number of drugs or adherence. However, appropriateness changes in duplication, effectiveness, dosage, and interactions criteria were observed in both groups, which can be explained by the Hawthorne effect. Costs have not been included in this study since they are part of a separate analysis.

This trial has several strengths, including the pragmatic, the population studied, and the rigorous methodological approach taken according to the standards recommended for cluster-randomized trials [53]. Losses to follow-up were small, less than 8% at the 12-month follow-up, which can be explained by the strong connection of patients to their primary-care physician in the Spanish National Health System. The losses in the intention-to-treat analysis were imputed by the multiple imputation. However, as stated in the trial protocol, we proposed to estimate using LOCF [25], which we considered to add a final sensitivity analysis (complete cases, ITT LOCF, and multiple imputed). No difference was found in the results, probably due to limited losses [54,55,56,57].

It was essential to have the FP and not the health center as a cluster, as the physician’s prescription was the dependent variable. In the Spanish National Health Service, patients are not registered with the practices but with a designated FP who is completely responsible for their care. Further, nurses do not prescribe and there is no independent prescriber pharmacist in practices. Several limitations could be identified: One limitation could be the FP contamination. To avoid this, FPs randomly assigned to the intervention group were asked to sign a confidentiality agreement and periodic reminders were sent until the end of the trial. Another limitation was that blinding was not possible, data collection was not blind. However, the main outcome was evaluated by three blinded family physician investigators with pharmacological expertise trained for the study. These evaluators had access to full patient medical records. The variability in clinical practice of the different participating physicians and their initial knowledge about the content that was developed in the training phase of the intervention also influenced the results.

Although we observed a significant difference across groups of the a priori (per protocol) defined magnitude, we acknowledge that a larger difference would provide more convincing evidence of the positive impact of the intervention. Although the intervention can be certainly considered to be at least promising, we cannot say the present evaluation provides conclusive evidence for a positive benefit. Future availability of empirical evidence of the clinical significance of MAI scores of the magnitude of the ones observed in the trial would increase our confidence in the interpretation of the benefit of the intervention.

## 5. Conclusions

The MULTIPAP intervention improved medication appropriateness, which was sustainable over the follow-up time. The small magnitude of the effect, however, advises caution in the interpretation of the results given the paucity of evidence for the clinical benefit of the observed change in MAI.

## Figures and Tables

**Figure 1 jpm-12-00752-f001:**
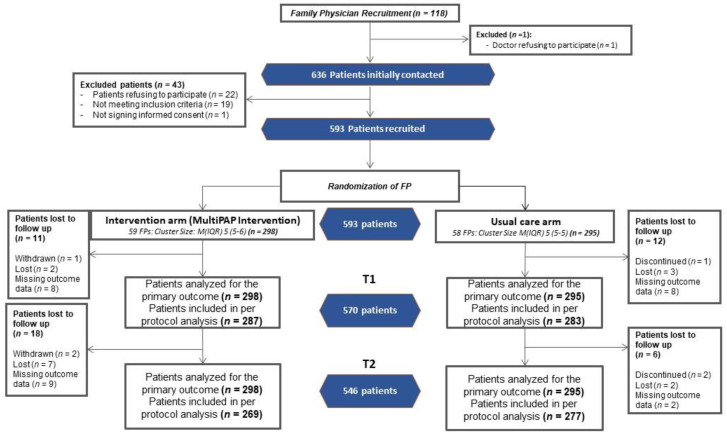
Study flow diagram.

**Table 1 jpm-12-00752-t001:** Baseline characteristics of FPs (clusters) and patients by study arm.

	Total	Control	Intervention
**Physician characteristics**	*n* = 117	*n* = 58	*n* = 59
Sex (female)	77 (65.8%)	41 (70.7%)	36 (61.0%)
Age of the professional, mean (SD)	52.1 (6.8)	52.4 (6.8)	51.8 (6.8)
Years in the professional position, mean (SD)	18.3 (3.4)	18.2 (3.7)	18.3 (3.0)
1–14 years	12 (10.3%)	4 (6.9%)	8 (13.6%)
15–19 years	27 (23.1%)	17 (29.3%)	10 (16.9%)
20 or more years	78 (66.7%)	37 (63.8%)	41 (69.5%)
Postgraduate tutor	75 (64.1%)	40 (69.0%)	35 (59.3%)
**Patient characteristics**	*n* = 593	*n* = 295	*n* = 298
Sex (Female)	331 (55.8%)	172 (58.3%)	159 (53.4%)
Age, mean (SD)	69.7 (2.7)	69.9 (2.7)	69.6 (2.7)
Spanish nationality	583 (98.3%)	290 (98.3%)	293 (98.3%)
Marital status			
Single	23 (3.9%)	12 (4.1%)	11 (3.7%)
Married	447 (75.4%)	224 (75.9%)	223 (74.8%)
Separated	29 (4.9%)	13 (4.4%)	16 (5.4%)
Widower	94 (15.9%)	46 (15.6%)	48 (16.1%)
Level of studies			
Did not complete primary studies	279 (47.0%)	142 (48.1%)	137 (46.0%)
Completed primary studies	196 (33.1%)	104 (35.3%)	92 (30.9%)
Bachelor or higher	118 (19.9%)	49 (16.6%)	69 (23.2%)
Social class *			
Supervisor, middle-management and director	234 (39.5%)	113 (38.3%)	121 (40.6%)
Skilled primary sector	217 (36.6%)	108 (36.6%)	109 (36.6%)
Unskilled	142 (23.9%)	74 (25.1%)	68 (22.8%)
Monthly income			
≤1050 euro	170 (28.7%)	88 (29.8%)	82 (27.5%)
1051–2250 euro	342 (57.7%)	163 (55.3%)	179 (60.1%)
≥2251 euro	59 (9.9%)	29 (9.8%)	30 (10.1%)
Unknown	22 (3.7%)	15 (5.1%)	7 (2.3%)
Home size (m^2^), mean (SD)	93.5 (42.9)	93.3 (48.6)	93.8 (36.4)
Number of cohabitants, mean (SD)	2.3 (0.7)	2.3 (0.6)	2.3 (0.7)
Functional Social support ^+^, mean (SD)	43.7(8.8)	43.7 (8.14)	43.6 (19.3)
Number of diseases, median (IQR)	5.0 (4.0, 7.0)	5.0 (4.0, 6.0)	5.0 (4.0, 7.0)
Number of drugs, median (IQR)	7.0 (6.0, 9.0)	7.0 (5.0, 8.0)	7.0 (6.0, 9.0)
Non-adherence (Medication Assessment Questionnaire)	242 (40.8%)	105 (35.6%)	137 (46.0%)
EuroQoL 5D-5 L, mean utilities (SD)	0.8 (0.2)	0.8 (0.2)	0.8 (0.2)
EuroQoL VAS, mean (SD)	65.5 (20.5)	65.2 (19.4)	65.9 (21.6)
Patient summated MAI score, mean (SD)	17.5 (16.8)	16.4 (14.6)	18.6 (18.6)

* Neoweberian occupational social class (CSO-SEE12). Gac Sanit. 2013, 27 (3): 263–272; ^+^ Duke-UNC 11.

**Table 2 jpm-12-00752-t002:** Outcome-adjusted differences for the primary (6-month follow-up) and secondary outcomes (12-month follow-up).

Primary Outcome (6 Months)	Control Group	Intervention Group	Adjusted Difference (95% CI)	*p*-Value
Difference in MAI (1) T1-T0 mean (SD) (*n*)	1.08 (0.41) (295)	3.43 (0.84) (298)	−2.42 * (from −4.27 to −0.59)	0.009
**Secondary outcomes (12 months) (1)**				
Difference in MAI T3-T0 mean (SD) (*n*)	1.19 (8.4) (277)	4.6 (11.1) (269)	−3.40 * (from −5.45 to −1.34)	0.001
Quality of life				
EQ-VAS (visual analogue scale) mean (SD) (*n*)	64.97 (19.75) (280)	68.18 (20.57) (272)	2.94 * (from −1.39 to 7.28)	0.18
EQ-5D-5 L (index) mean (SD) (*n*)	0.780 (0.182) (280)	0.763 (0.213) (272)	−0.006 * (from −0.034 to 0.022)	0.68
Treatment adherence				
Medication Assessment Questionnaire	71/280 (25.4%)	73/272 (26.8%)	−0.048 ^‡^ (from −0.65 to 0.56)	0.87
Medication safety				
Absolute incidence of adverse drug reactions	1 (1–1) (291)	1 (1–2) (290)	0.49 ^†^ (from −0.12 to 1.11)	0.11
Patient perception of shared decision making				
NHS questionnaire (3)	259/275 (94.2%)	250/264 (94.7%)	0.09 ^‡^ (from −0.91 to 1.10)	0.85
CollaboRATE (5)	218/275 (79.3%)	211/264 (79.9%)	0.03 ^‡^ (from −0.67 to 0.74)	0.92
Use of health services				
Hospital admissions (IQR) (*n*)	1 (1–2) (280)	1 (0–1) (272)	−0.14 ^†^ (from −0.57 to 0.30)	0.52
Visits to emergency services (IQR) (*n*)	1 (1–2) (280)	1 (1–3) (272)	0.18 ^†^ (from −0.06 to 0.41)	0.14
Number of FP consultations (*n*)	7 (4–10) (280)	7 (4–11) (272)	0.07 ^†^ (from −0.11 to 0.25)	0.44
Number of primary-care nurse consultations (*n*)	4 (2–7) (280)	4 (2–8) (272)	0.10 ^†^ (from −0.15 to 0.35)	0.43

MAI, Medication Appropriateness Index. (1) Secondary outcomes at 6 months in Appendix A; (3) ordinal variable dichotomized for ease of presentation; (5) dichotomous CollaboRATE score variable (i.e., top score for all three questions or not top score). * Beta-coefficients, analyses are adjusted multilevel linear regression. ^†^ Coefficients, analyses are adjusted Poisson multilevel regression. ^‡^ Coefficients, analyses are adjusted logistic multilevel regression.

**Table 3 jpm-12-00752-t003:** Factors associated with changes in the difference in MAI score (from baseline to 6-month follow-up).

	Coefficients	95% CI	*p* Value
Receiving intervention (yes)	1.78	0.29 to 3.29	<0.001
Baseline summated MAI Score	0.29	0.25 to 0.33	0.02
Living alone (yes)	−1.59	−3.08 to −0.11	0.03
Functional social support *	0.07	0.01 to 0.14	0.03
Number of drugs 5–6 Drugs	ref		
7–9 Drugs	−1.44	−2.76 to −0.12	0.03
≥10 Drugs	−0.94	−2.76 to 0.86	0.30

References categories omitted. Goodness-of-fit, Akaike’s information criterion (AIC) 3893.965; Bayesian information criterion (BIC) 3937.422; empty model intraclass correlation coefficient 0.19; final model ICC 0.15; * Duke-UNC scale.

## Data Availability

The data that support the findings of this study will be available from research unit uinvestigación.ap@salud.madrid.org, but restrictions apply to the availability of these data, which were used under license for the current study, and so are not publicly available. However, each new project based on these data must be previously submitted to CEICA for approval. However, data are available from the authors upon reasonable request and with permission of the project’s principal investigators Alexandra Prados-Torres at sprados.iacs@aragon.es, Daniel Prados-Torres at uand.prados.sspa@juntadeandalucia.es, and Isabel Del Cura at isabel.cura@salud.madrid.org.

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
