# Peer review of "How to Improve Healthcare for Patients with Multimorbidity and Polypharmacy in Primary Care: A Pragmatic Cluster-Randomized Clinical Trial of the MULTIPAP Intervention"

_jpm, 2022, doi:10.3390/jpm12050752_

Round 1

Reviewer 1 Report

In the current study as primary objective the authors evaluated the effectiveness of the complex MULTIPAP intervention, which implements the Ariadne principles in a primary care population of young-elderly patients with multimorbidity and polypharmacy, at improving the appropriateness of prescription. As secondary objectives they evaluated the effects of the MULTIPAP intervention on quality of life, use of health services, adherence to treatment and medication safety.

Some remarks:

-line 140: is primary objective. Add please primary

-point 2.1.: please add the number of patients who were included in the study and the number of family physicians which were recruited

-line 170, you wrote that the patients were recruited from December 2016 to Jamuary 2017.  Why the results of the study will be published only in 2022? No changes can occur in 5 years? -I think it would be better to move the information presented at point 2.5 Outcomes to point 2.1. Methods  and to delete the subtitle outcomes -You underlined the several strenghts of the trial. Please underline also all the weaknesses of the trial

- At the end of the article please formulate a more elaborate conclusion 

In my opinion the trial is well designed and presented. Also, the subject is of great interest. So, I recommend publishing the article in Journal of Personalized Medicine after Minor revisions

Author Response

Thank you very much for the revision and the suggestions. We answer every suggestion point-by-point.

line 140:  is primary objective. Add please primary.  suggestion  has been included. 

-point 2.1.: please add the number of patients who were included in the study and the number of family physicians which were recruited. This data is into the result section  and the study flow diagram acording to the recomendation of the control cluster statement. 

-line 170, you wrote that the patients were recruited from December 2016 to Jamuary 2017.  Why the results of the study will be published only in 2022? No changes can occur in 5 years?

After the last visit of the last patient followed ( Dec 2018), in 2019 we started the data analysis and the writting of the manuscript and in earlier 2020 we started the editorial process. This situation added  to our clinical work during the pandemic slowed the procces. We consider that there were not change in practice since 2016-2017 affecting the findings on current practice.  Furthermore, the current pandemic is affecting the monitoring and control of chronically ill patients by prioritizing COVID care, making these findings even more relevant.

 -I think it would be better to move the information presented at point 2.5 Outcomes to point 2.1. Methods  and to delete the subtitle outcomes 

We have followed the exact CONSORT 2010 checklist of information to include when reporting a cluster randomised trial where it recomends and specific subsection after the intervention. 

-You underlined the several strenghts of the trial. Please underline also all the weaknesses of the trial. 

Thank you for your suggestion, although limitations and weaknesses were described and the end of the discusion they were not propperly labelled. Your suggestion  has been included in the discussion section after the strenghts in lines 464 and from 466 to 478. 

At the end of the article please formulate a more elaborate conclusion 

Your suggestion  has been included from line 479 to 483

English language and style has been check by the American Journal Expert ( plesase see the atachment) .

Reviewer 2 Report

This is a well-described and executed study. The results of the study will contribute to the existing body of knowledge. In addition, it was interesting and valuable to observe non-significant results. I have only a minor suggestion to add p-values to Table 1 that I believed would improve the manuscript.

Author Response

Thank you very much for your evaluation.

Regarding your suggestion to add p-values to Table 1, we have this data but it has not been included acording to the CONSORT CLUSTER STATEMENT  where it´s said " Random assignment by individual ensures that any differences in group characteristics at baseline are the result of chance rather than some systematic bias." (Schulz K, Chalmers I, Hayes R, Altman DG. Empirical evidence of bias: dimensions of methodological quality associated with estimates of treatment effects in controlled trials. JAMA 1995;273:408-12) as  It can be seen in the given example ( table 3). 

Please let us knwon if the editorial deccision for this journal  is to add p-values so we could include in the final table 1. 

English language and style has been check by the American Journal Expert ( plesase see the atachment).
